# Magmatic connectivity among six Galápagos volcanoes revealed by satellite geodesy

Eoin Reddin [1] ✉, Susanna K. Ebmeier [1] ✉, Eleonora Rivalta [2,3], Marco Bagnardi [4,8], Scott Baker[5], Andrew F. Bell [6], Patricia Mothes[7] & Santiago Aguaiza[7]

Shallow magmatic reservoirs that produce measurable volcanic surface deformation are often considered as discrete independent systems. However, petrological analyses of erupted products suggest that these may be the shallowest expression of extensive, heterogeneous magmatic systems that we show may be interconnected. We analyse time series of satellite-radar-measured displacements at Western Galápagos volcanoes from 2017 to 2022 and revisit historical displacements. We demonstrate that these volcanoes consistently experience correlated displacements during periods of heightened magma supply to the shallow crust. We rule out changes in static stress, shallow hydraulic connections, and data processing and analysis artefacts. We propose that episodic surges of magma into interconnected magmatic systems affect neighbouring volcanoes, simultaneously causing correlations in volcanic uplift and subsidence. While expected to occur globally, such processes are uniquely observable at the dense cluster of Western Galápagos volcanoes, thanks to the high rate of surface displacements and the wealth of geodetic measurements.

Volcanic unrest and eruption may be initiated by processes at the Earth's surface[1,2] or by changes in deeper supplies of magma[3,4]. At oceanic hotspots, thin crust promotes the frequent effusive eruption of basaltic lavas that rapidly flush through crustal magmatic systems[5] or accumulate at shallow depths (e.g., <2 km from the surface[6]). This thin oceanic crust facilitates the rapid ascent of magma through multiple trans-crustal pathways instead of through a few established magmatic storage zones in thicker crust[7]. Therefore, oceanic hotspots with multiple sub-aerial volcanoes (Fig. 1i) are excellent sites to identify and study rapid magma migration through connected magmatic systems. At the Western Galápagos islands of Isabela and Fernandina (Ecuador), there are six active volcanoes atop the oceanic crust with a maximum thickness of 18 km[8]. This high spatial density of volcanoes,

alongside the high-magnitude deformation and eruption rates routinely observed there, make the islands an excellent site to understand the relationship between volcanic deformation, magma flux, and the underlying trans-crustal magmatic systems.

The six major volcanic centres of the Western Galápagos (Alcedo, Cerro Azul, Darwin, Fernandina, Sierra Negra, and Wolf (Fig. 1)) have their magma supplied by the Galápagos plume and are each separated from their neighbour by lateral distances of only 30–45 km. These volcanoes are underlain by vertically extensive magmatic systems, with discrete levels of magma storage[9,10]. Magma supply through each system varies according to the maturity of the volcano[9,10]. The shallowest magmatic storage zones lie in the upper 4 km of the crust[6,11–13], where volume changes produce displacement

[1]School of Earth and Environment, University of Leeds, Leeds LS29JT, UK. [2]Department of Physics and Astronomy, Alma Mater Studiorum, University of Bologna, Viale Berti Pichat 8, Bologna 40126, Italy. [3]Helmholtz Centre Potsdam, GFZ German Research Centre for Geosciences, Telegrafenberg, Potsdam 14473, Germany. [4]Cryospheric Sciences Laboratory, NASA Goddard Space Flight Center, Greenbelt, MD, USA. [5]BOS Technologies LLC, Lafayette, CO, USA. [6]School of GeoSciences, University of Edinburgh, James Hutton Road, Edinburgh EH9 3FE, UK. [7]Instituto Geofísico de la Escuela Politécnica Nacional, Ladrón de Guevara, E11-253 Quito, Ecuador. [8]Present address: U.S. Geological Survey, Volcano Science Center, Vancouver, WA, USA. ✉e-mail: eeer@leeds.ac.uk; s.k.ebmeier@leeds.ac.uk

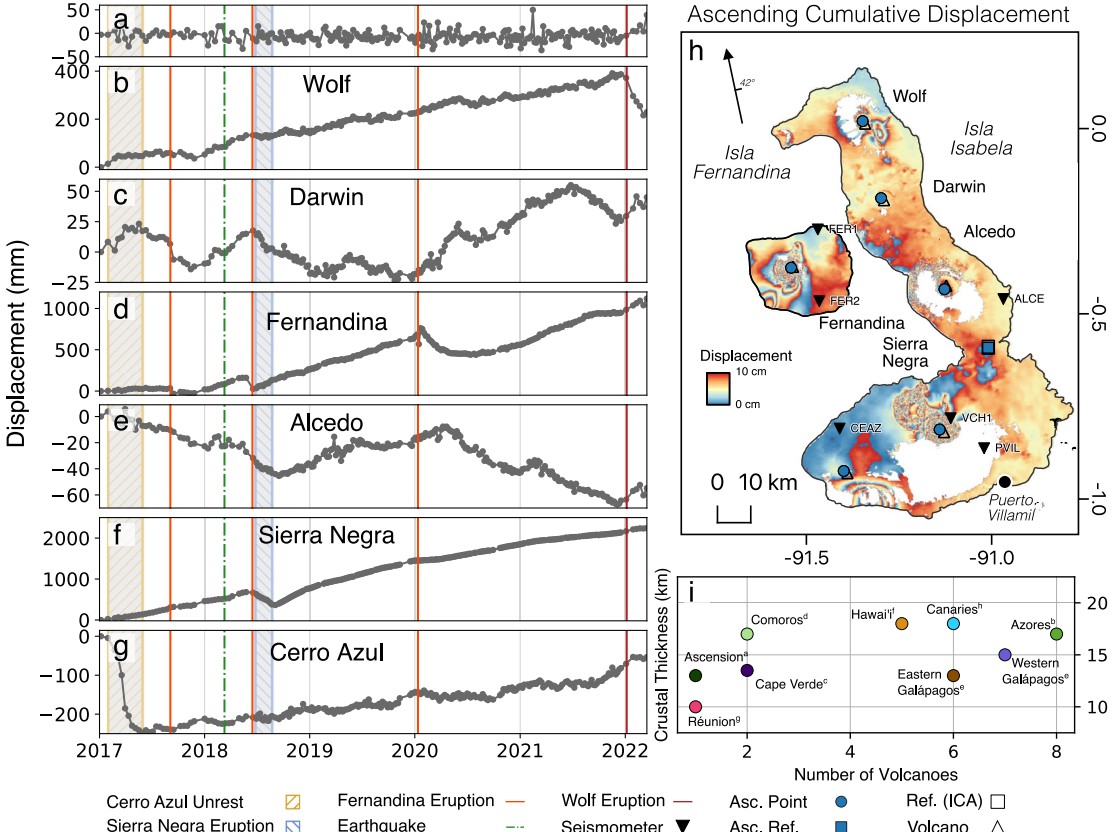

**Fig. 1 | Ascending Sentinel-1 cumulative displacement time series for the Western Galápagos, from 06/01/2017 to 17/03/2022. a** Time series of average displacement of the reference pixel (annotated by the blue square in panel (**h**)). **b**–**g** Displacement time series at each of the major volcanoes of the Western Galápagos. Periods of significant unrest, including unrest at Cerro Azul and eruptions at Fernandina, Sierra Negra, and Wolf, are annotated. Displacement values may differ from previous studies due to the choice of point plotted. For example, Sierra Negra subsided by a maximum of 8 m during the 2018 eruption[6], resulting in phase-decorrelation in our dataset. Therefore, we plot a point that may not capture this maximum displacement in order to prevent time series gaps. **h** Wrapped cumulative displacement map of the Western Galápagos across the entire time series (2017–2022). Each fringe corresponds to 10 cm of displacement in the satellite line-of-sight direction. The arrow shows the satellite heading, as well as the average incidence angle. The annotated points refer to pixels used during correlation analysis, while the reference area is used during Independent Component Analysis (Fig. 2). **i** Crustal thickness and number of volcanoes at oceanic hotspots. References for crustal thicknesses are as follows: a[49], b[50], c[51], d[52], e, f[53], g[54], h[55]. Source data are provided as a Source Data file.

patterns that are easily measured by satellite radar thanks to high phase coherence[14,15].

Here, we use satellite radar data to produce time series of volcanic displacement and show that there are correlations between deformation at all six volcanoes during various periods of eruption and quiescence and that these correlations are especially strong during episodes of elevated magma flux.

## Results

### Shared displacements in InSAR time series

Using Interferometric Synthetic Aperture Radar (InSAR), we construct displacement time series for Isla Fernandina and Isla Isabela from January 2017 to March 2022. These time series ("Methods") capture displacements associated with major unrest episodes (e.g., Cerro Azul, 2017) as well as five eruptions (Fernandina in 2017, 2018, and 2020, Sierra Negra, 2018, and Wolf, 2022). There are obvious associations between displacement trends before and after eruptions (Fig. 1 and Supplementary Fig. S1). For example, Darwin began to subside as Fernandina and Sierra Negra erupted in 2018. After these eruptions, Alcedo switched to uplift at the same time as Sierra Negra resumed inflation. Additionally, Fernandina, Cerro Azul, and Darwin all showed simultaneous changes in displacement direction and rate in mid-2021, approximately 3 months prior to the Wolf 2022 eruption. Cross-correlation analysis[16] showed that time lag is close to zero (Supplementary Fig. S2) at the temporal resolution of our InSAR observations,

implying strong vertical connectivity in magmatic systems or that any differences in ascent rate to reach the interconnected magmatic zones are short relative to InSAR acquisition spacing. This is consistent with earlier observations of simultaneous unrest at each of Fernandina, Wolf and Alcedo (Supplementary Fig. S3). Fernandina and Alcedo showed changes in surface displacement during a sequence of local earthquakes (each <M5.0) between 2006 and 2007[17]. During the April 2009 eruption of Fernandina, Alcedo switched from uplift to subsidence[17], while the uplift rate at Wolf slowed. This ended 10 years of steady uplift at Wolf (Supplementary Fig. S3). The uplift rate at Fernandina decayed during the sudden deflation at Alcedo in May 2010[17]. Similarly, GPS data show that the rate of inflation at Sierra Negra slowed during the May 2008 eruption of Cerro Azul[17]. Some sparse petrological data also hints at magmatic connectivity here, as isotopically similar magmas have erupted at adjacent volcanoes[18]. However, these similar samples are rare; there is evidence that Galápagos volcanoes have been erupting magmas of distinct compositions for the last 10 Ka, each sampling a distinct part of a geochemically heterogeneous plume[19].

The increased temporal density of interferometric data provided by the European Spaces Agency's Sentinel-1 satellite relative to previous SAR sensors allows a more systematic approach to testing the relationships between time series. We use three independent methods to do this ("Methods"): identification of turning or inflection points in time series (Fig. 1), calculation of correlation coefficients between time

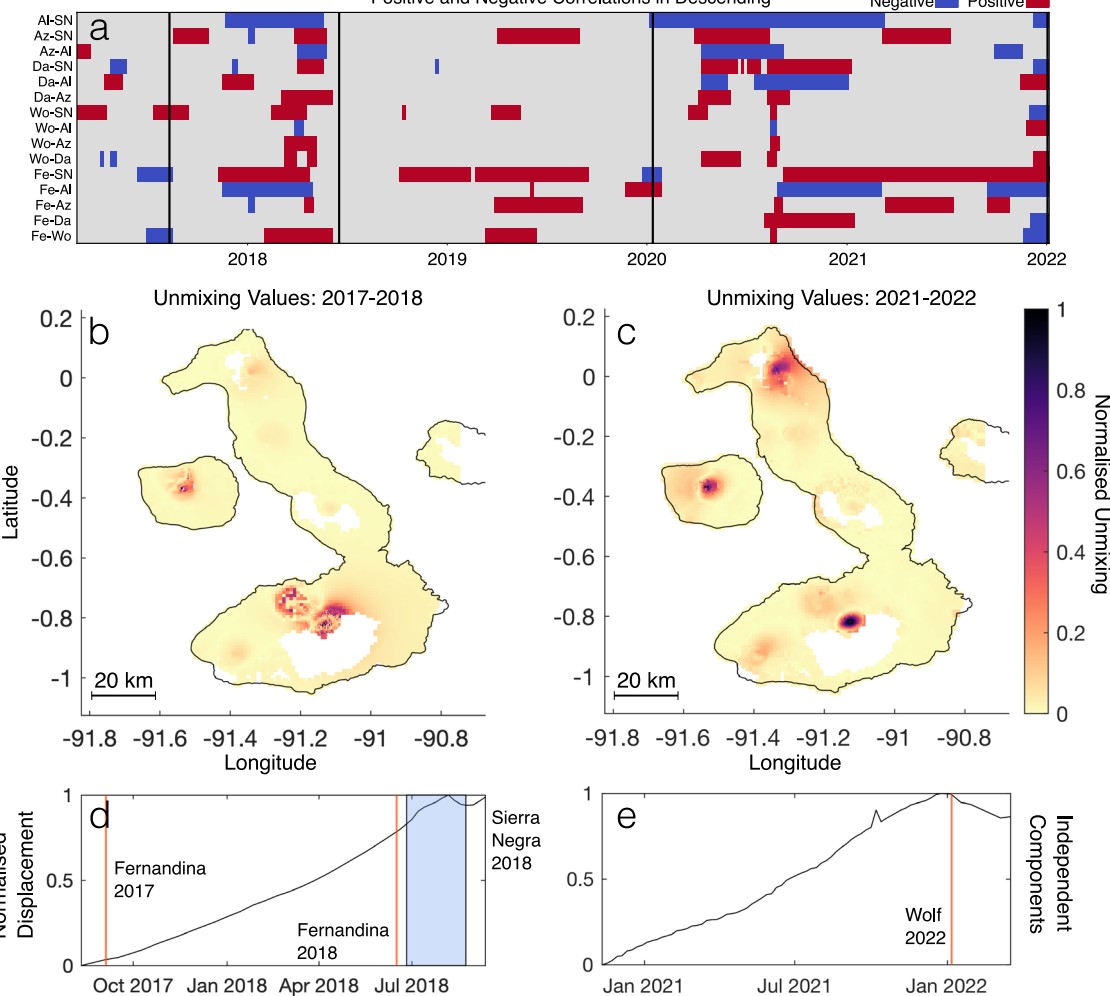

**Fig. 2 | Results of Independent Component Analysis, and comparison with correlation analysis, on Descending Data (ICA results in both track directions for all periods can be found in Supplementary Figs. S4 and S5). a** Results of correlation analysis for each pair of volcano time series, in descending track direction. Grid colour indicates if the correlation is positive (red), or negative (blue), and >|0.9|. Source data are provided as a Source Data file. **b** Spatially reconstructed unmixing values for the corresponding independent component (panel **d**) for the Fernandina eruptions in 2017, 2018, and the Sierra Negra eruption in 2018, from 11/08/2017 to 17/09/2018. The colour indicates the relative strength of the unmixing value. **c** Spatially reconstructed unmixing values for the corresponding independent component (panel **e**), prior to the 2022 eruption of Wolf, from 11/11/2020 to 30/03/2022. **d** Independent component corresponding to the unmixing values in panel (**b**). Displacement is normalised on the *y*-axis. **e** Independent component corresponding to the unmixing values in panel (**c**). Note that the independent component does not necessarily mirror the shape of the original time series. For example, the contribution of the independent component shown in panel e results in rate changes at Wolf, Sierra Negra, Fernandina and Cerro Azul (Supplementary Fig. S1).

series pairs using a rolling windowed approach (Fig. 3a), as well as temporal Independent Component Analysis (ICA), as a robust test of the statistical independence of signals (Fig. 2). In combination, these approaches demonstrate that correlated displacements occur persistently between all pairs of volcanoes in the Western Galápagos, and are not only associated with eruptions (spanning days–weeks) but also episodes of inter-eruptive unrest (weeks–months). This suggests a high degree of complex, inter-magmatic system connectivity in the Western Galápagos.

Fernandina, Wolf, Alcedo, Cerro Azul, and Sierra Negra showed correlated displacement multiple times during several unrest episodes between 2006 and 2011[17]. We observe similar patterns in the Sentinel-1 time series, particularly before and during the 2018 eruptions of Fernandina and Sierra Negra and preceding the 2022 eruption of Wolf (Fig. 1). Figure 2 shows independent temporal components and their weighting, derived for both of these phases of activity. Between 2017 and 2018, we retrieved an independent component of deformation associated with shared uplift. This component is strongest at Fernandina and Sierra Negra and also present at Wolf. Similarly, correlation

analysis shows that this period had the highest number of strongly correlated time series of the entire study period (57% of possible time series combinations had a correlation coefficient of >|0.9|) (Fig. 3a, b). When applied to time series during the 2022 eruption of Wolf, as well as the preceding year, we retrieve a single independent component that describes deformation at each of Wolf, Fernandina, and Sierra Negra. Correlation analysis shows that this period had the second-highest number of strongly correlated time series, with 53% of all possible time series combinations having a correlation coefficient of >|0.9|. In these two periods, visual inspection of the time series, Independent Component Analysis, and correlation analysis all indicate that Galápagos volcanoes are deforming in a correlated manner. From late 2020 to 2021, Darwin uplifted as Alcedo began to subside. Though we do not clearly retrieve an independent component for this event (Supplementary Figs. S4 and S5), we find that 43% of possible time series combinations show strongly correlated deformation, the third highest number of the study period. These periods have the highest number of strong correlations of the entire time series, showing clear changes in displacement behaviour at multiple volcanoes, with two of

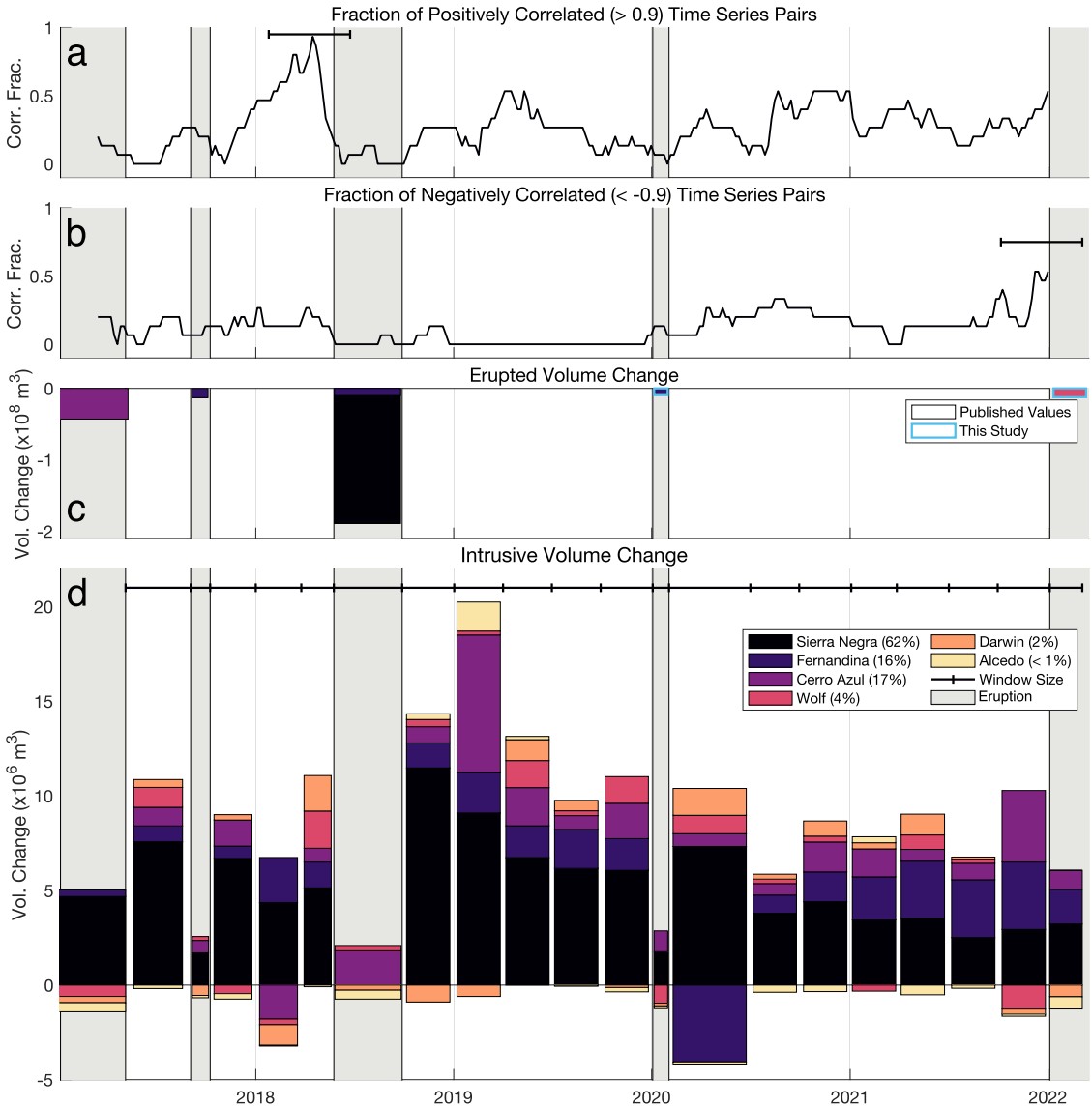

**Fig. 3 | Change in the number of strong pair-wise correlation coefficients with time, as well as volume flux into each Western Galápagos volcano. a** Number of strong positive correlations between each pair of time series determined using windowed correlation analysis ("Methods"). The correlation coefficient is calculated between each pair of volcanoes (Supplementary Fig. S10), and the number of values that are >0.9 are counted. The count for each window is plotted at the centre of the window (5 months), the size of which is annotated by the horizontal black bar. **b** Number of strong negative correlations between each pair of time series, determined using windowed correlation analysis ("Methods"). The correlation coefficient is calculated between each pair of volcanoes (Supplementary Fig. S10), and the number of values that are <−0.9 are counted. Source data are provided as a Source Data file. **c** Estimates of volume flux for each unrest period, either from published analysis or estimated here. **d** Volume change with time at each Western Galápagos volcano for a sill source. The spatial coordinates (X, Y, and Z values) of a sill are determined for each volcano and then held constant, while the variables that control volume change (length, width, and opening) are allowed to vary. The width of each bar corresponds to the length of the period modelled, as does the horizontal black line, while different colours represent each volcano. Periods of significant unrest (e.g., eruptions) have been masked out (vertical grey bars) for the corresponding volcano (e.g., Sierra Negra 2018). In the Galápagos, eruptions typically comprise multiple sources at different levels in the crust that cannot be accurately modelled by a single, fixed source.

them showing shared independent components between multiple volcanoes. Most of the correlations between pairs of displacement time series are positive (Fig. 3a and Supplementary Fig. S10), indicating that volcanoes were deforming in the same direction (e.g., both inflating). This agrees with the observation of inter-eruptive inflation at each volcano, as multiple monotonic processes will correlate with one another. We identify change points in the time series on the basis of displacement direction or rate and assume a Poisson distribution for the number of changes, to estimate the probability of two volcanoes with random rate changes having correlated turning points. We identify an average of 4 change points in each volcanic deformation time series (Supplementary Figs. S7 and S8) and find the probability of just

two volcanoes randomly experiencing a major change in deformation rate in the same month to be 0.21%, and even lower for correlated changes at >2 volcanoes. Similarly, using eruption rates since the early 20th century[20], we find that the probability of Fernandina and Sierra Negra erupting in the same 6-month period (as occurred in 2005 and 2018) would be 0.35% if eruptions were randomly spaced in time. Therefore, we conclude that Galápagos volcanoes routinely deform and even erupt in a correlated manner.

While coupled volcanic unrest has been reported globally[21–24], quantitative proof of relationships between markers of unrest, such as displacements, are rare[3,16,17] and were previously limited by a scarcity of ground-based infrastructure and the lower temporal resolution of

satellite geodetic measurements (e.g., ERS-1/2, Envisat, and ALOS-1 had repeat times varying from 35 to 46 days, while Sentinel-1 has a repeat time of 6 days when in a two satellite constellation). Observations from the Galápagos are unique in that they demonstrate temporally consistent correlated displacements due to changes in shallow magmatic system pressures over multiple decades (as determined by semi-continuous space geodetic coverage between 2006 and 2011, and 2017 and 2022), including during inter-eruptive periods. We rule out the possibility that the correlations in our time series are InSAR measurement artefacts by systematically testing the impact of reference pixel selection (Fig. 1a, Supplementary Fig. S6). We also test the impact of island-wide atmospheric phase contributions by making corrections from weather models and assessing the seasonality of our time series ("Methods").

### Interpreting correlated deformation

Observations of correlated deformation and eruptions at other volcanic centres have been attributed variously to the impact of tectonic or magmatic static stress changes[25–27], hydraulic magmatic connections between volcanic systems[16,23,24], or pore-pressure diffusion[28]. No recorded earthquakes or magmatic intrusions that occurred during our observation period were sufficient to cause such persistent correlations in displacement. A seismic station on Sierra Negra's caldera rim (VCH1) recorded no earthquakes greater than magnitude 5.4 between 2012 and 2022, with the largest events due to slip along intra-caldera faults preceding and during eruptions[29,30]. Among all eruptions that occurred during this study, the largest was the 2018 eruption at Sierra Negra (Fig. 3c). The erupted volume here was $147 \pm 71 \times 10^6$ m$^3$ DRE (at least 14 times more than that of the 2017 eruption of Fernandina $(9.7 \pm 4.9 \times 10^6$ m$^3$ DRE)[31]. However, even the pressure drop caused by this eruption only produced significant dilational strain (e.g., $>4 \times 10^{-5}$) within 10 km of the volcano (Supplementary Fig. S9), and therefore not sufficient to explain correlated deformation seen at Fernandina and Wolf. This leaves the possibility that dynamic stress changes from a magmatic intrusion at a location unobservable from InSAR or the existing seismic network (Fig. 1) (e.g., below the seafloor between Isabela and Fernandina) could cause similar changes in shallow reservoirs. However, the lack of a simple spatial relationship between the locations where correlated deformation occurs (Fig. 2), and the switching between episodes of positive and negative correlation (Fig. 3a, b), are inconsistent with such a mechanism.

There is also no evidence for extensive, persistent shallow hydraulic connections between Western Galápagos volcanoes. Vertical magma migration in the shallow crust occurs during both unrest and eruption[11] and is more consistent with distinct stacked sub-volcanic systems than lateral magma movements. Shallow reservoirs at <4 km lie above deeper storage zones at >5 km[10] at Fernandina[5,10,11,32], Sierra Negra[6] and Wolf[5,10,13]. The lateral migration of magma through shallow sills is a common phenomenon in the Galápagos, and it produces high-magnitude deformation when it does occur (e.g., Cerro Azul, 2017[14], Sierra Negra, 2018[6,33] (Fig. 1h), and Fernandina 2006, and 2007[11]). However, the extensive episodes of positively correlated deformation that we observe during inter-eruptive periods do not show measurable deformation associated with shallow magma movements. In Hawai'i, like the Western Galápagos, Mauna Loa and Kīlauea show correlated deformation yet erupt isotopically distinct magmas[3,27,28]. An astheno-spheric melt layer is proposed to facilitate coupling between these volcanoes across distances of >34 km[28]. Pore-pressure diffusion through this layer affects the magma supply rate to the shallow reservoirs and allows coupled deformation between the volcanoes while maintaining their isotopic heterogeneity[28]. Such a layer has been seismically imaged at Hawai'i—a collection of sills between 36 and 43 km depth, with seismicity indicating magmatic pathways to both Mauna Loa and Kīlauea[34], while extensive magma storage in the upper mantle has also been observed in the Canary Islands, at El Hierro[35]. A

similar structure in the Galápagos may reconcile our observations of correlated deformation with the isotopic evidence of distinct magmas between each volcano[19].

This structure could exist at depths beyond geodetic detection, at the base of the crust (ranging from 12 km at Fernandina to 18 km at Sierra Negra[8]). We consider the simplest explanation for correlations in Galápagos displacements to be temporally variable melt supply from the plume into an asthenospheric melt layer that causes neighbouring sub-volcanic reservoirs to simultaneously pressurise via pore-pressure diffusion. A variable melt supply would explain our observations of correlations in shallow magmatic reservoir pressure that are dominantly positive but with some modifications due to (1) the impact of eruptions and resulting shallow stress changes on individual sub-volcanic systems, and (2) variations in partitioning of melt entering the shallow crust due to differences in ascent mechanisms (Fig. 4).

There are multiple strands of evidence for a time-varying melt flux into the Western Galápagos. We know that magma flux into the shallower crust there is periodic[36,37], and that historical eruptions have occurred in clusters (e.g., there were pauses between Galápagos eruptions from 1998 to 2005, and 2009 to 2015). There have also been discrete episodes of exceptionally high rate uplift (e.g., Sierra Negra 2005–2018[6,14]). Periods of elevated magma flux may result in magmatic 'flushing' (the rapid migration of primitive magmas through the sub-volcanic system, at Fernandina and Wolf)[5], or the resurgence (repressurisation of a formerly contracting or inactive intrusion) of quiescent systems over months to years (Alcedo, 2007 (Supplementary Fig. S3); Darwin, 2020, (Fig. 1)[36,37]. We compare correlation coefficients over windows of 25 SAR acquisitions (approximately 5 months) with published volumetric estimates[14,31] and with estimates of intrusive volume flux at each volcano between 2017 and 2022 ("Methods"). Our estimates of intrusive magma flux (Figs. 3 and 4) use inversions of InSAR-derived displacements spanning either an eruption or inter-eruptive windows of 3 months ("Methods"). We observe that the periods with the highest number of strong correlations between pairs of time series occur during an eruption (e.g., Fernandina, Sierra Negra, 2018, and Wolf, 2022) or when there is an increase in magma supply (e.g., as Darwin uplifted in 2020) (Fig. 1). For example, Cerro Azul, Fernandina, Sierra Negra, and Wolf show only positively correlated deformation in 2019 (Fig. 2a, Supplementary Fig. S10), corresponding with the greatest intrusive volume flux of the entire time series $(1.9 \times 10^7$ m$^3$ from January to March, 2019). Similarly, a period of positive volume flux to all volcanoes in late 2020 (Fig. 3) resulted in a high number of positive correlations (Fig. 3a). This supply increase was observed on the surface with the uplift of Darwin, the highest magnitude uplift measured at the volcano throughout our time series (Fig. 1). This observation of correlated displacement occurring during episodes of increased magmatic flux agrees with observations from between 2007 and 2011[17]. Then, heightened magma flux, evidenced by Alcedo's resurgence[37], caused similar displacements at five volcanoes. This demonstrates that the correlated unrest discussed here, from 2006 and 2022, captures a representative rather than an anomalous period of Galápagos volcanism.

The majority of positive correlations is consistent with a common response to variation in melt supply from a deeper source. The positive correlation at Cerro Azul, Fernandina, Sierra Negra, and Wolf in 2019 (Fig. 2a) and at Cerro Azul, Darwin, Fernandina, Sierra Negra, and Wolf in late 2020 (Fig. 2a) occurred during a period of increased melt supply to each volcano (Fig. 3d), and as Darwin resurged (Fig. 1). This supply increase caused pressurisation of shallow reservoirs across the Galápagos resulting in correlated uplift (Fig. 1). During periods of heightened supply, magma ascends either slowly via porous flow, or rapidly through dike propagation[38]. Episodes of positively correlated subsidence were more unusual (e.g., Fernandina–Sierra Negra, 2018) and shorter in duration but can similarly be attributed to simultaneous eruptions or a drop in magma supply, allowing cooling processes such

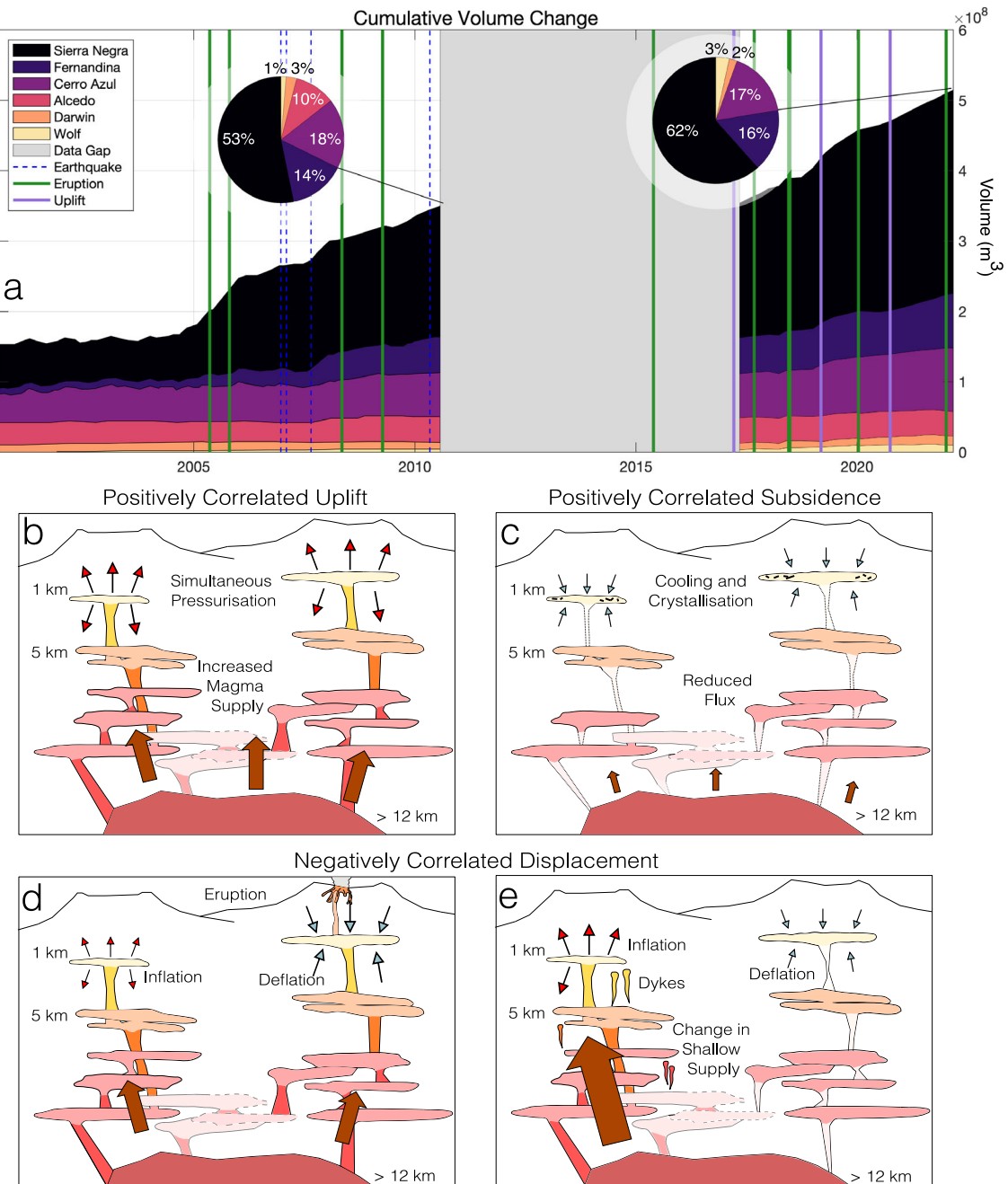

Fig. 4 | **Mechanism of correlated displacements in the Western Galápagos.**
**a** Cumulative intrusive volume in the Western Galápagos from 2000 to 2022
(modified from ref. 56), with periods of correlated displacements annotated (this
study, and modified from ref. 17). Each coloured area corresponds to a different
volcanic system, while the coloured vertical bars denote the observed evidence of
related displacement (e.g., during an eruption, or volcanic unrest)—eruptions that
occurred, but did not result in observed related displacements are faintly marked.
The grey area marks periods where there was a lack of enough satellite data to
perform modelling ("Methods"). The embedded pie charts show the cumulative
distribution of magma at each volcano, first for the period from 2000 to 2010
(modified from ref. 56) and second for the from 2017 to 2022 (i.e., not including
volumes from 2000 to 2010). Alcedo is not included as it subsided over this period,
and neither are volumes lost during an eruption (e.g., those that were removed in

Fig. 3). **b** Schematic diagram of a proposed mechanism of correlated uplift.
Neighbouring volcanic systems are supplied from the same deeper source; periodic
changes in magma supply from this source cause these volcanoes to uplift simul-
taneously. These systems may be hydraulically connected at some depth to facil-
itate correlated displacements during an eruption. **c** Schematic diagram of a
proposed mechanism of correlated subsidence. As magma supply wanes, cooling
and crystallisation dominate, decreasing reservoir volume and allowing neigh-
bouring volcanic systems to deform simultaneously. **d** Schematic diagram of a
proposed mechanism of negatively correlated unrest. One of these then erupts
while the other continues to inflate, causing subsidence, while the neighbour
continues to inflate. **e** Negatively correlated displacement due to a change in
magma supply.

crystallisation to dominate the shallowest parts of the magmatic sys-
tem. For example, subsidence at both Alcedo (1997–2001[39]) and Sierra
Negra (2000–2002[40]) may be due to the cooling of intruded products
during a period of decreased magma supply.

Negative correlations may also be explained by pore-pressure
diffusion, where increased magma supply to one volcano creates a
pressure gradient throughout the asthenospheric melt layer. For
example, in the 6 months prior to the 2022 eruption of Wolf, the uplift

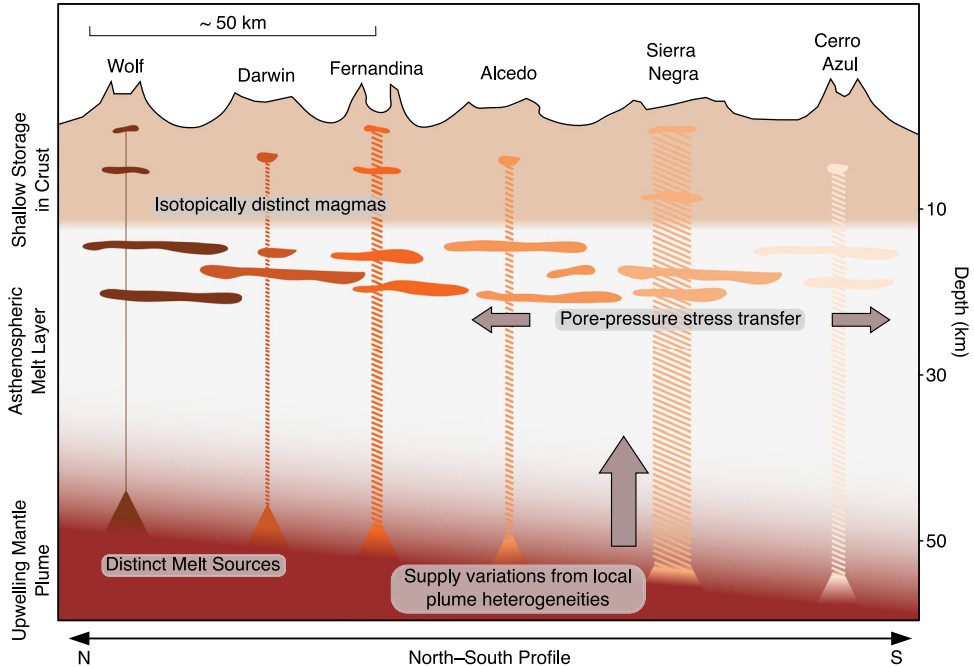

**Fig. 5 | Schematic illustration of our proposed bottom-up mechanism for correlated deformation at Galápagos volcanoes.** The recent melt flux to each volcano (Fig. 3) is represented by the width of hatched areas, with Sierra Negra accounting for an average of 55% of the total supply since 2000 (Fig. 4). Each volcano samples a geochemically distinct area of the Galápagos plume, where local compositional variations control supply[42], though there may be occasional mingling between heterogeneous magma batches[18]. Connectivity occurs at the base of the crust through pore-pressure stress transfer between geochemically distinct sills[28,34]. The flux of magma to the shallow crust, causing measurable deformation, varies in magnitude and partitioning according to a combination of plume supply variations and eruption, as illustrated in Fig. 4.

rate and time series turning points were correlated between multiple volcanoes, while deformation trends were anti-correlated. Specifically, displacements in the descending satellite track at Cerro Azul (at −0.900 E, −91.356 N) increased to an average of 0.1 m yr$^{-1}$ from 0.04 m yr$^{-1}$ in the previous 6 months (Supplementary Fig. S1). This was accompanied by subsidence at Darwin (Fig. 1) and a decrease in the uplift rate at Sierra Negra (Fig. 1).

Our modelling of shallow intrusive flux into a sub-volcanic sill (Fig. 3d) shows that from 2021, magma supply to Cerro Azul steadily increased while steadily decreasing at Sierra Negra. This negative correlation was also seen during the 2008 eruption of Cerro Azul when the inflation rate at Sierra Negra slowed[17]. The GPS station GV02 was closest to the centre of the shallow reservoir at Sierra Negra and showed the greatest change in vertical displacement of all Sierra Negra stations during the eruption of Cerro Azul, again indicating a magmatic response[17]. Another explanation for negatively correlated deformation is the transient response of a magmatic system to eruption when stress drops in shallow reservoirs temporarily alter the rate of magma ascent from background values in a 'top-down' process[41].

## Conceptual model for common magmatic systems

Our conceptual model of volcanism in the Western Galápagos (Figs. 4 and 5) is of distinct, vertically extensive sub-volcanic magmatic zones that all respond to dynamic magma supply from the Galápagos plume, analogous to Hawai'i. This dynamic supply may cause dynamic pore-pressure changes throughout an asthenospheric melt layer, varying supply to the shallow crust. This melt layer is likely composed of geochemically distinct sills, allowing dynamic stress transfer while preserving isotopic heterogeneity between volcanoes (Fig. 5). Melt flux from the plume varies temporally, flushing through different sub-volcanic systems during eruptions[5], allowing subsiding volcanoes to resurge[37], and creating periods of elevated eruptive activity (e.g., 2005–2009, 2015–2022). However, magma is not supplied equally to each Galápagos volcano, with Sierra Negra receiving >50% of the total supply from 2000 to 2022 (Fig. 4). Subsidence (indicative of limited magma supply to the shallow crust) has been observed twice at Sierra Negra, between 2000 and 2002[40], and 2011 and 2012[6], in both instances during a quiescent period of decreased eruptive activity across the Western Galápagos. At Kīlauea, local compositional heterogeneities at the plume alter the degree of partial melting, affecting magma supply to the shallow crust and the eruption rate[42]. Therefore, we speculate that variations in magma supply at Sierra Negra, caused by local plume heterogeneities, are primarily responsible for driving dynamic stress through an interconnected asthenospheric sill complex. An aspect that remains unclear is the timescales across which this process occurs and how well they are resolved by Sentinel-1 InSAR (return period of 6–12 days). Future modelling studies are required to fully investigate this. Supply variations to other Galápagos volcanoes will also affect pore-pressure stresses, though to a lesser extent than at Sierra Negra, as will major eruptions or morphological changes at specific volcanoes, causing deviations from these general trends.

The tight grouping of active, deforming volcanoes in the Western Galápagos makes their connectivity uniquely observable using geodetic data. However, multi-volcano cycles of unrest and eruption driven by a common source are likely to exist globally, as has been discussed in Hawai'i.

We speculate that the thin crust beneath the Western Galápagos (and potentially the Azores (Fig. 1i)) is underlain by a melt layer comprising geochemically distinct sills, each sampling a different part of the Galápagos plume, and allowing stress interactions at the point of supply to sub-volcanic magmatic zones (Fig. 5). This promotes multiple ascent pathways within close proximity and feeds six actively deforming Galápagos volcanoes within 100 km of one another. We anticipate that at other ocean hotspot systems, such as Ascension, Cape Verde, Reunion, or Comoros (Fig. 1i), magma flux is also ultimately driven by deep melt supply, but that fewer active volcanic centres inhibit observations of direct connections between surface displacements and deeper magmatic source zones. With this study, we

show that volcanic displacement in the Western Galápagos and beyond should be viewed as a window into deeper, interconnected trans-crustal magmatic systems.

# Methods

## InSAR

We use approximately 2000 interferograms spanning January 2017 (the first date from which regular satellite acquisitions were available in both track directions), until March 2022, towards the culmination of the 2022 eruption of Wolf Volcano (Fig. 1). These interferograms are automatically constructed using the LiCSAR Sentinel-1 InSAR Processor[43]. This is a fully automated chain for processing Sentinel-1 InSAR data[43]. Image pairs have a minimum temporal spacing of between 6 or 12 days when S1-A and S1-B are in constellation and are geocoded to pixel spacing of approximately 100 m. The network and perpendicular baselines are presented in Supplementary Fig. S11. As these interferograms are automatically produced, we manually perform a quality check and then re-make those interferograms containing gaps or unwrapping errors. We use the Generic Atmospheric Correction Online Service (GACOS)[44] to minimise the error introduced by tropospheric noise. This correction reduced the standard deviation of the unwrapped phase in the descending track, from a dataset mean of 3.2 radians to 2.7 (Supplementary Fig. S12). However, the standard deviation slightly increased in the ascending track, from 4.12 radians to 4.27 (Supplementary Fig. S12). Here, maps of tropospheric delay are created using data from the European Centre for Medium-Range Weather Forecast, then subtracted from the LiCSAR-generated interferograms. Finally, we use these corrected interferograms to construct a time series of displacement using the LiCSBAS time series inversion software[45]. This is an open-source software package designed for time series analysis of LiCSAR products using a small baseline inversion. LiCSBAS utilises a phase closure technique to identify and remove bad interferograms, such as those with unwrapping errors[45]. The final time series are filtered using a Gaussian kernel, with a spatial width of 2 km and a temporal width of 24 days (three times the average acquisition spacing).

## Correlation analysis

We perform rolling correlation analysis to identify periods of correlated displacements between representative time series for each volcano. For this analysis, and for the later Independent Component Analysis, we use time series previously produced using LiCSBAS. The primary criteria for the time series selected for analysis are that they must be representative of displacements at the caldera across the time series and that the time series contains no data gaps (e.g., due to decorrelation during an eruption), as such gaps may present as a false positive or negative correlation. The locations of the time series selected for presentation can be found in Fig. 1 and Supplementary Fig. S1. We then discretise time series into temporal windows, where each window is offset by one acquisition from the previous window. To ensure that each window has consistent temporal coverage, we linearly interpolate each time series, keeping a constant separation of 6 days between each data point. We use windows of 150 days or approximately 5 months; eruptions in the Western Galápagos frequently last months (e.g., in 2018, Sierra Negra erupted for 2 months, while in 2022, Wolf erupted for 5 months). With this window size, we ensure that we can compare multiple windows across entire eruptions, as well as any immediate pre- and post-eruptive correlation. We also ensure that we are comparing time series where the source of deformation is magmatic rather than shorter-term processes such as brittle failure and slip-along trapdoor faults. We test the impact of window size and find that windows of between 90 and 150 days produce consistent correlations. An example of using a 3-month window can be seen in Supplementary Fig. S13. We then calculate the correlation coefficient for each window to retrieve a rolling correlation coefficient, pair-wise, for each set of volcanic time series. The correlation coefficient is a measure of the strength of the linear relationship between two variables, given from a range of −1 to 1. Correlation increases closer to either −1 or 1, while values closer to zero are more independent. The sign indicates whether the values are positively or negatively correlated. The correlation coefficient of two random variables, A and B ($\rho_{AB}$), be expressed through Equation (1).

$$\rho_{AB} = \frac{\text{cov}(A,B)}{\sigma_A \sigma_B} \qquad (1)$$

Where $cov$ is the covariance of A and B, and $\sigma$ is their respective standard deviation. We identify strongly correlated displacement periods between volcanoes when absolute windowed correlation coefficients are >0.9. We visualise these periods in Fig. 3, which represents the number of these strong correlations at each date. We plot both the sum of the absolute correlation coefficients and the non-absolute coefficients as a means of visualising when positive or negative correlations are active.

## Independent component analysis

We select episodes of high correlation for further analysis using Independent Component Analysis (ICA) (e.g.[46]). ICA is a blind source separation technique that is used to retrieve independent sources from a mixed signal. ICA is based on the principle that a mixed signal, X, is composed of a linear mixture of multiple components, S (X = AS), where A is a mixing matrix. Using the fastICA algorithm[47], we iterate to find an unmixing matrix, W, the inverse of A, such that S = WX, and the non-Gaussinity of W is maximised. An initial random value for W is selected before iteration. Therefore, global convergence is not assured, and the retrieved independent components may vary between each run. For each selected time interval, we perform ICA using approximately 8000 time series across the island (a total of 16,000 points). To ensure reasonable computation times, we downsample the number of time series based on the radial distance to each caldera centre (Supplementary Fig. S14) on the assumption that the majority of displacement occurs within the caldera. This step provides three levels of resolution; the caldera floor is downsampled to 0.4 of its original resolution, and the caldera flanks to 0.25. Once downsampled, we use a k-means algorithm to identify independent components that appear consistently between different iterations, ensuring we converge on all W values. For each selected time period (as identified through correlation analysis), we run fastICA 100 times and save the output. We identify unique independent components by calculating the sum of the squares of each one returned and cluster similar time series using this value, using k-means. We calculate the mean independent component and unmixing value for each cluster. We visualise where these Independent components are active by plotting the absolute of each W value with its corresponding latitude and longitude (e.g., Fig. 2). With this approach, we can accurately reconstruct known displacement patterns (e.g., Cerro Azul 2017 and Sierra Negra 2018), and robustly identify the volcanoes that are affected by the same independent component. In addition to Fig. 2, more results of this analysis can be found in the Supplementary material.

## Estimation of intrusive flux

We estimate the volume flux with time into a shallow sill[11,13] at each volcano from 2017 to 2022. While Descending data is available from 14/11/2015, Ascending is only available from 06/01/2017. As we perform a joint inversion, we use this latter date as a starting point. To do this, we perform a Bayesian inversion for best-fit source parameters using the MATLAB-based Geodetic Bayesian Inversion Software (GBIS)[15]. We first crop the spatial extent of the displacement map such that only one volcano is considered in each inversion. We then pick a representative time period for each volcano and model the optimal source, using a

joint inversion of the Ascending and Descending tracks. This representative period is selected such that there are no eruptions at the volcano and that the observed displacement is from inflation or deflation of the sub-volcanic reservoir. We then slice our cumulative displacement time series into approximately 3-month windows. However, this window size is not constant during periods of significant unrest; windows are cropped to include all of an unrest period, such that the effect of volume loss only occurs in a single modelled window. This results in 21 windows of cumulative displacement for each volcano, or 126 total windows to be modelled. For each of these, we characterise the spatial variance using a semi-variogram[15] and ensure that the degree of quadtree downsampling is similar in both track directions. When slicing the cumulative displacement windows in either track direction, the dates do not exactly overlap, as the Descending acquisition typically occurs the day after the Ascending. Therefore, the inputs for our joint inversion are typically offset by one day, though we assume that interim displacement is negligible compared to the 3-month period studied. Once the optimal source has been constrained for each volcano, we hold the X, Y and Z coordinates constant and allow the opening, length, and width of the sill to vary for each window. The data, model, residual, and retrieved parameters can be found in Supplementary Figs. S15–S20. In each case, we verify that the retrieved depth of the sill agrees with what has been determined by previous studies, if available. We then calculate the associated volume change to determine the change in volume with time (e.g., Figs. 3 and 4). Though we model volume change in the best-fit source during eruptions and unrest (Cerro Azul, 2017, Fernandina, 2017, 2018, 2020, Sierra Negra, 2018, and Wolf, 2022), these volumes may not agree with previous observations. Many sources may be active during a volcanic eruption, making volume change estimates unreliable for our single source. To deal with this, draw on published results for eruptive episodes and use single sources for the inter-eruptive periods. Finally, the dilational change during an eruption, as presented in Supplementary Fig. S9, was modelled with Coulomb 3.1[48]. We modelled this using a point source at Sierra Negra, with a volume change of $-2 \times 10^8$ m$^3$[31].

## Data availability

The Interferometric Synthetic Aperture Radar data used here were processed using the LiCSAR interferometric processor and are freely available on the COMET-LiCS Sentinel-1 InSAR portal (https://comet. nerc.ac.uk/COMET-LiCS-portal/). The GACOS atmospheric correction data is freely available and can be found at http://www.gacos.net. The processed time series and modelling data are available at https://doi. org/10.6084/m9.figshare.23661363; source time series data are provided with this paper. Source data are provided with this paper.

## Code availability

The LiCSBAS time series analysis software is available at https://github. com/yumorishita/LiCSBAS. GBIS is available at https://comet.nerc.ac. uk/gbis/. fastICA is available at http://research.ics.aalto.fi/ica/fastica/. Code relating to correlation analysis and Independent Component Analysis can be found at https://doi.org/10.6084/m9.figshare. 23661363.

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

## Acknowledgements

This work is supported by the National Environmental Research Council (NERC) of the United Kingdom through the British Geological Survey Centre for the Observation and Modelling of Earthquakes, Volcanoes, and Tectonics (NERC-BGS COMET, https://comet.nerc.ac.uk). Co-authors are supported as follows: E.Red. was supported by a European Space Agency Living Planet Fellowship originally awarded to S.K.E.; NERC Independent Research Fellowship (NE/R015546/1), S.K.E.; German Research Foundation (DFG), Grant N. 634756, RI 2782/2, E.Riv.; NSFGEO-NERC grant on Sierra Negra NE/W007274/1, A.B.

## Author contributions

E.Red. performed time series analysis of InSAR data and performed the analysis of these time series, and modelling of magma flux. E.Red. and

S.K.E. co-wrote the manuscript. E.Riv., M.B., S.B., A.F.B., P.M. and S.A. provided critical feedback.

## Competing interests

The authors declare no competing interests.
