## [Peer Review File · Nature Communications]

Magmatic connectivity among six Galápagos volcanoes revealed by satellite geodesyREVIEWER COMMENTS

Reviewer #1 (Remarks to the Author):

This is an interesting paper showing correlated ground deformation among the western Galapagos volcanoes. The deformation signals are significant and clearly not due to systematic errors. The result is surprising given the distance between many of these volcanoes. The authors rule out, correctly I believe, elastic interactions or shallow plumbing system connections as the explanation. They suggest wide-spread surges in magma supply to the deeper parts of the respective volcanoes.

I think this is plausible, but feel that a more complete investigation of the significance of the correlations is warranted. In particular, active volcanoes tend to inflate between eruptions. In most cases they deflate only during eruptions (modulo deformation due to shallow hydrothermal processes). Thus, they are much more likely to be inflating at any given time than deflating. Note that any monotonic trend will correlate with another monotonic trend, and the correlation will be positive if they are all inflating. This simply reflects mantle supply to all of the volcanoes. Thus, the correlated positive trends, for example from essentially the end of the 2018 Sierra Negra eruption to the start of the 2020 Fernandina eruption, visible at all volcanoes except Darwin and noted in the text, is not really that surprising.

What is more interesting and noteworthy is the correlation of higher temporal frequency deformation signals. For example, Alcedo and Darwin reverse trend just before the 2022 Wolf eruption. The question then is, is this statistically significant, given that Alcedo and Darwin experience more trend reversals? I suspect that answer is yes, but important to nail down. One could address that by looking at the number of trend reversals and ask what the probability of these occurring together randomly are. Another example might be the trend reversal at Darwin coincident with the 2020 Fernandina eruption. Given the frequent trend reversals at Darwin, this may or may not be significant.

For that matter, what are the odds of nearly simultaneous eruptions at Sierra Negra and Fernandina in 2018?

In summary, this is a significant paper and worthy of publication. I think it could be made stronger by focusing on the shorter term fluctuations and clearly showing that these correlations are significant.

Paul Segall

Minor points:

Figure 1. Wouldn't it be better if the time series were roughly N to S rather than alphabetical?

I find Figure 2 c) and e) confusing. During the period 2021 - 2022 this figure shows Wolff correlated with SN and Fernandina. The latter have positive trend, but no reversal at time of Wolff eruption. I rather see a negative correlation between Wolff and Alcedo during this time. Possibly a positive correlation with Cerro Azul. What am I missing?

line 131 "Additionally, Fernandina, Cerro Azul, and Darwin all showed simultaneous changes in displacement direction and rate prior to the Wolf 2022 eruption." Did you mean Alcedo rather than Cerro Azul?

Figure 4. When you say cumulative volume, does this mean cumulative intrusive volume? Not diminished by eruption?

Reviewer #2 (Remarks to the Author):

This is one of the best papers I've ever read. It definitely belongs in either Nature or Nature Communications. The discovery that these volcanoes are hydraulically connected in the mantle is a fundamental aspect of both volcanology and tectonism, a finding that will change conceptual models of volcanic regions beyond the Galápagos and eventually lead to compelling mechanical models. The fact that this connectivity was discovered by satellite geodesy is mind-boggling to me.

I am not an expert in InSAR data acquisition and processing, but I have complete trust in the authors' capabilities: they are leaders in the field. The statistical methods, especially the correlation of temporal events, seem to be robust, but again this is not my specialty.

I have made a number of detailed comments on the manuscript.

I have a few quibbles:

1. Figure 4 (a summative schematic) shows only 2 volcanoes. One of the most surprising results is that pressure is transmitted across volcanoes at a regional scale. So I would like to see several volcanoes (perhaps a N-SW transect) in a regional cross section, illustrating the authors' vision of the distribution and transport of melt.

2. This idea was brought up to me about 15 years ago by Falk Amelung, who thought there might feedback between neighboring Cerro Azul and Sierra Negra. I was resistant to the idea, because the compositions of lavas from the western Galapagos volcanoes are individually distinguishable: if you give me a lava, I can tell you where it is from on the basis of mantle geochemical indicators (e.g. Ba/Nb, La/Nb and Sr-, Nd-, and Pb- isotopes). So the pressurized melt zone in the mantle must keep its compositional zoning, despite regional melt being present and fluctuating. Despite the fluctuations in melt supply, there is no evidence that the compositions of the magmas change. Although a thorough analysis of geochemistry is beyond the scope of this paper, it might be worth bringing up these two observations and their implications.

3. There are other lessons from Hawaii: see the 2023 Science paper from Wilding, which is relevant to this work. Also, several papers on Kilauea by Pietruszka, which use geochemical evidence to identify decadal pulsations in magma supply by the plume. So far as I know, these effects have not been discovered in Galapagos, either because they don't exist or the data are insufficient.

4. The discoveries in this paper really beg for regional mechanical and fluid dynamic models of the feedbacks between the deep interconnected magma supply, the transcrustal plumbing systems, and the subcaldera pre-eruptive storage areas. That is certainly beyond the scope of this article, but I look forward to someone tackling it!

Review of manuscript no. NCOMMS-23-05124-T

Magmatic Connectivity among Six Galapagos Volcanoes Revealed by Satellite Geodesy

E. Reddin, S. Ebmeier, E. Rivalta, M. Bagnardi, S. Baker, A. Bell, P. Mothes, and S. Aguaiza

Summary:

Reddin et al. present deformation timeseries for a number of volcanic systems in the Galapagos. They identify periods of correlated (both positive and negative) deformation between the different volcanoes that they assign to the presence of an interconnected trans-crustal magma system that responds to periods of heightened magma supply from the underlying plume, as well as volcanic eruptions. As a petrologist, I don't feel entirely qualified to evaluate and review the methods and/or results obtained by the authors, so I will focus instead on their discussion and the implications this has from a petrological/geochemical standpoint.

I believe that there is one aspect that needs further discussion, specifically how an interconnected magmatic system can be reconciled with the abundant petrological evidence for almost completely chemically isolated magmatic reservoirs? This point is discussed in more detail below and, once addressed, I'd see no reason from a geochemical and/or petrological stand point to delay publication.

Geochemical isolation:

The authors indicate, in the abstract and elsewhere, that there is petrological analysis that supports the presence of interconnected magmatic systems. This statement is largely supported by the presence of a single study, Geist et al. 1999 (in *Geology*). The data in this paper indicates that maybe ~1% of flows/eruptions from the Galapagos volcanoes have geochemical compositions that are more similar to those usually found at a different volcano. However, in my opinion, what is most striking about this paper, and the many other papers that have been published on the western Galapagos volcanic systems by Geist and co-workers, is that in most cases, the Galapagos volcanoes are geochemically (and petrologically) distinct, and do not freely share magma or show any distinct signs that they are connected at depth.

In figure 4 of this study, the authors present a model where there is a large magma body at depth feeding different sub-volcanic systems. Can the authors comment on the nature of this body, different regions of which must be fed from different mantle source components. Do you believe that this is a truly connected magma body that can share magma freely between systems (seemingly implausible with the geochemistry), is it an interconnected mush zone with limited lateral transport of melt, or is it a series of geochemically distinct sills, each of which might supply a different volcanic structure (this idea might have interest given the recent seismic observations of deep sill networks between Mauna Loa and Kilauea on Hawai'i; <https://doi.org/10.1126/science.ade5755>). On a similar note, each volcano samples a different portion of the Galapagos mantle plume, would we necessarily expect there to be synchronous magma surges across this geochemically variable plume structure?

In summary, I'd like to see a greater discussion on how the correlated deformation signals can be reconciled with the fact that petrological evidence indicates that these systems are mostly isolated.

We thank the reviewers for their insightful comments, responding to these comments has given us new perspective on the nature of Galápagos correlations. We have addressed each comment as set out below, and can also be seen in colour highlights in the revised manuscript.

Reviewer #1 (Remarks to the Author):

1. *This is an interesting paper showing correlated ground deformation among the western Galapagos volcanoes. The deformation signals are significant and clearly not due to systematic errors. The result is surprising given the distance between many of these volcanoes. The authors rule out, correctly I believe, elastic interactions or shallow plumbing system connections as the explanation. They suggest wide-spread surges in magma supply to the deeper parts of the respective volcanoes. I think this is plausible, but feel that a more complete investigation of the significance of the correlations is warranted. In particular, active volcanoes tend to inflate between eruptions. In most cases they deflate only during eruptions (modulo deformation due to shallow hydrothermal processes). Thus, they are much more likely to be inflating at any given time than deflating. Note that any monotonic trend will correlate with another monotonic trend, and the correlation will be positive if they are all inflating. This simply reflects mantle supply to all of the volcanoes. Thus, the correlated positive trends, for example from essentially the end of the 2018 Sierra Negra eruption to the start of the 2020 Fernandina eruption, visible at all volcanoes except Darwin and noted in the text, is not really that surprising.*
 - o Agreed, we discuss the significance of short-term correlations in bullet point 2. And note the point about correlating monotonic trends in text:
 - o Line 233 (Now Line 241): “This agrees with the observation of inter-eruptive inflation at each volcano, as multiple monotonic processes will correlate with one another.”
2. *What is more interesting and noteworthy is the correlation of higher temporal frequency deformation signals. For example, Alcedo and Darwin reverse trend just before the 2022 Wolf eruption. The question then is, is this statistically significant, given that Alcedo and Darwin experience more trend reversals? I suspect that answer is yes, but important to nail down. One could address that by looking at the number of trend reversals and ask what the probability of these occurring together randomly are. Another example might be the trend reversal at Darwin coincident with the 2020 Fernandina eruption. Given the frequent trend reversals at Darwin, this may or may not be significant. For that matter, what are the odds of nearly simultaneous eruptions at Sierra Negra and Fernandina in 2018?*

We have addressed this comment by adding in some further analysis of correlations in the higher temporal frequency deformation signals. We take the reviewers suggestion of focusing on the numbers of trend reversals and demonstrate statistical significance as follows:

- o Line 231 (Now Line 243): “We use a change point algorithm and assume a Poisson distribution for the number of rate changes, to estimate the probability of two volcanoes with random rate changes having correlated turning points (Section 6). We identify an average of 5 change points in each volcanic deformation time series (Figures S7, S8), and find the probability of just two volcanoes randomly experiencing a major change in deformation rate in the same month to be 0.3%, and even lower for correlated changes at >2 volcanoes. Similarly, using eruption rates since the early 20th century we find that the probability of Fernandina and Sierra Negra erupting in the same 6 month period (as occurred in 2005, and 2018) would be 0.3% if eruptions were randomly spaced in time.”
- o Line 233 (Now Line 254): “and even erupt, in a correlated manner.” “~~in a correlated manner rather than being an exceptional event associated only with particular eruptions.~~”
- o Line 616 (Now Line 724): “Change Point Analysis
As a test of statistical significance, we calculate the probability of simultaneous changes in deformation randomly occurring at multiple volcanoes by assuming a Poisson distribution (Equation 6), where X is the random variable, k is the number of times an event occurs and λ is the average number of times the event occurs (e.g. the number of turning points in a time series).

$$P(X = k) = \frac{e^{-\lambda} \lambda^k}{k!}$$

To estimate λ, we fit segmented linear regression lines to our data, so that each segment is separated by a change point (PELT off-line change point detection algorithm, in the ruptures Python library [56]). The number and location of change points is determined using a cost function, alongside a penalty value to avoid over-fitting (Figures S7, S8). We take the average number of change points across all time series as λ=5, and use this to estimate the probability of multiple volcanoes showing changes in deformation rate at the same time. Similarly, we estimate the probability of eruptions randomly occurring at the same time using Equation 6, using average eruption rates at each volcano between 1911–2023 (1911 was the first recorded 20th eruption).”

- o Included Supplementary Figures S7 and S8

3. *Figure 1. Wouldn't it be better if the time series were roughly N to S rather than alphabetical?*

We have changed the time series order as suggested.

4. *I find Figure 2 c) and e) confusing. During the period 2021 - 2022 this figure shows Wolff correlated with SN and Fernandina. The latter have positive trend, but no reversal at time of Wolff eruption. I rather see a negative correlation between Wolff and Alcedo during this time. Possibly a positive correlation with Cerro Azul. What am I missing?*
- Line 220: We have added in clarification to the figure caption as follows:
"Note that the independent component does not necessarily mirror the shape of the original time series. For example, the contribution of the independent component shown in e) results in rate changes at Wolf, Sierra Negra, Fernandina and Cerro Azul (Figure S1)"
5. line 131 "Additionally, Fernandina, Cerro Azul, and Darwin all showed simultaneous changes in displacement direction and rate prior to the Wolf 2022 eruption." Did you mean Alcedo rather than Cerro Azul?
- We have edited to clarify this:
(Now Line 134): " Fernandina, Cerro Azul, and Darwin all showed changes in displacement direction and rate in mid-2021, approximately 3 months prior to the Wolf 2022 eruption"
6. *Figure 4. When you say cumulative volume, does this mean cumulative intrusive volume? Not diminished by eruption?*
- Line 487 (Now Line 534): Yes – "Cumulative intrusive volume" added for clarity

Reviewer #2 (Remarks to the Author):

We thank reviewer #2 for their very helpful comments. The paper suggestions were particularly useful in helping us to refine our conceptual mechanism of correlated deformation.

In-email comments:

7. *Figure 4 (a summative schematic) shows only 2 volcanoes. One of the most surprising results is that pressure is transmitted across volcanoes at a regional scale. So I would like to see several volcanoes (perhaps a N-SW transect) in a regional cross section, illustrating the authors' vision of the distribution and transport of melt.*
- Figure 5 has now been included, showing a schematic of melt distribution and transport. Please find it included at the bottom of this document. The caption for the figure is as follows:

“Schematic illustration of our proposed bottom-up mechanism for correlated deformation at Galapagos volcanoes. The recent melt flux to each volcano (Figure 3) is represented by the width of hatched areas, with Sierra Negra accounting for an average of 55% of total supply since 2000 (Figure 4). Each volcano samples a geochemically distinct area of the Galápagos plume, where local compositional variations control supply [50], though there may be occasional mingling between heterogeneous magma batches [25]. Connectivity occurs at the base of the crust through pore pressure stress transfer between geochemically distinct sills [34, 42]. Flux of magma to the shallow crust, causing measurable deformation, varies in magnitude and partitioning according to a combination of plume supply variations and eruption, as illustrated in Figure 4.”
8. *This idea was brought up to me about 15 years ago by Falk Amelung, who thought there might feedback between neighboring Cerro Azul and Sierra Negra. I was resistant to the idea, because the compositions of lavas from the western Galapagos volcanoes are individually distinguishable: if you give me a lava, I can tell you where it is from on the basis of mantle geochemical indicators (e.g. Ba/Nb, La/Nb and Sr-, Nd-, and Pb- isotopes). So the pressurized melt zone in the mantle must keep its compositional zoning, despite regional melt being present and fluctuating. Despite the fluctuations in melt supply, there is no evidence that the compositions of the magmas change. Although a thorough analysis of geochemistry is beyond the scope of this paper, it might be worth bringing up these two observations and their implications.*
- We have addressed this point thoroughly in our new submission. Please refer to responses to reviewer #3 for a line by line breakdown of changes
9. *There are other lessons from Hawaii: see the 2023 Science paper from Wilding, which is relevant to this work. Also, several papers on Kilauea by Pietruszka, which use geochemical evidence to identify decadal pulsations in magma supply by the plume. So far as I know, these effects have not been discovered in Galapagos, either because they don't exist or the data are insufficient.*
- We take the points about similarities with Hawai'i, alongside the points about the distinct magma compositions, and refer to them directly in the new illustration of our conceptual model presented in Figure 5. Please refer to response to reviewer 3 for a line-by-line breakdown of changes. I would like to thank the reviewer for the papers suggested here, they were particularly useful.
10. *The discoveries in this paper really beg for regional mechanical and fluid dynamic models of the feedbacks between the deep interconnected magma supply, the transcrustal plumbing systems, and the subcaldera pre-eruptive storage areas. That is certainly beyond the scope of this article, but I look forward to someone tackling it!*
- Agreed! The new seismic observations by Wilding and the previous pore pressure model by Gonnermann present a compelling case for similar studies in the Galápagos.

(Comments in PDF File):

11. *Line 68: Need reference for crustal thickness:*
- Added Feighner and Richards 1994 here (Now Line 70)
12. *Line 388: It would seem to me that this is the most fundamental first-order observation, so maybe it should go in the lead.*
- Moved to the lead
 - Line 388 (Now Line 434): **The majority of strong correlations between pairs of displacement time series are positive (Figure 3a and Figure S10)** The majority of positive correlations
 - Line 231 (Now Line 238): **Most of the correlations between pairs of displacement time series are positive (Figure 3a and Figure S10), indicating that volcanoes were deforming in the same direction (e.g. both inflating).**
13. *Line 407-409: This seems weird, since the volcanoes are widely separated.*

- We have reordered these statements for clarity, emphasising that changes in partitioning would affect all volcanoes (e.g., Figures 4 and 5), not specifically Wolf and Cerro Azul as a pair:
- (Now Line 455): **the descending satellite track** and time series turning points were correlated between multiple volcanoes, while deformation trends were anti-correlated. Specifically, displacements in the descending satellite track,

Reviewer #3 (Remarks to the Author):

1. *The authors indicate, in the abstract and elsewhere, that there is petrological analysis that supports the presence of interconnected magmatic systems. This statement is largely supported by the presence of a single study, Geist et al. 1999 (in Geology). The data in this paper indicates that maybe ~1% of flows/eruptions from the Galapagos volcanoes have geochemical compositions that are more similar to those usually found at a different volcano. However, in my opinion, what is most striking about this paper, and the many other papers that have been published on the western Galapagos volcanic systems by Geist and co-workers, is that in most cases, the Galapagos volcanoes are geochemically (and petrologically) distinct, and do not freely share magma or show any distinct signs that they are connected at depth.*

We thank the reviewer for this very helpful comment. The isotopic heterogeneity at Galápagos volcanoes is a fundamental observation, which we did not consider in our initial submission. We have edited our manuscript to include the bigger petrological picture as follows.

- o Line 41-42: ~~potentially interconnected magmatic systems~~ - magmatic systems, that we show may be interconnected
- o Line 147-148 (Now Line 151): ~~Petrological data also indicate magmatic connectivity between Galápagos systems; isotopically similar magmas have erupted at adjacent volcanoes [25].~~ Some sparse petrological data also hints at magmatic connectivity here, as isotopically similar magmas have erupted at adjacent volcanoes [25]. However, these similar samples are rare; there is evidence that Galapagos volcanoes have been erupting magmas of distinct compositions for the last 10 Ka, each sampling a distinct part of a geochemically heterogenous plume [26].
- o Line 276 (Now Line 297): ~~with triggering from either static or dynamic stresses.~~ with such a mechanism.
- o Line 289-293 (Now Line 315): ~~Although stress transfer due to porous flow through an extensive mush zone can conceivably produce coupled unrest over distances of < 10 km [27, 34, 40, 41], this could not account for the extended episodes of correlated displacements across distances of up to 100 km — the distance between Cerro Azul and Wolf.~~ In Hawai'i, like the Western Galápagos, Mauna Loa and Kīlauea show correlated deformation, yet erupt isotopically distinct magmas [3, 33, 34]. An asthenospheric melt layer is proposed to facilitate coupling between these volcanoes, across distances of > 34 km [34]. Pore-pressure diffusion through this layer affects magma supply rate to the shallow reservoirs, and allows coupled deformation between the volcanoes, while maintaining their isotopic heterogeneity [34]. Such a layer has been seismically imaged at Hawai'i, a collection of sills between 36–43 km depth, with seismicity indicating magmatic pathways to both Mauna Loa and Kīlauea [42], while extensive magma storage in the upper mantle has also been observed in the Canary Islands, at El Hierro [43]. A similar structure in the Galápagos may reconcile our observations of correlated deformation with the isotopic evidence of distinct magmas between each volcano [26].
- o Line 295-306 (Now Line 330): ~~This leaves us with the possibility that connections exist between sub-volcanic systems at depths beyond geodetic detection, either due to the limited diameter of the islands, or because magma movement through deeper, hotter crust does not result in elastic deformation. At the Galápagos crustal thickness ranges from 12 km at Fernandina, to 18 km at Sierra Negra [19]. We consider the simplest explanation for correlations in Galápagos displacements to be temporally variable melt supply from the plume, that causes neighbouring sub-volcanic reservoirs to simultaneously pressurise. A variable melt supply would explain our observations of correlations in the shallow magmatic reservoir pressure that are dominantly positive, but with some modifications due to (1) any changes in partitioning of melt flux between volcanoes and (2) the impact of eruptions and resulting shallow stress changes on individual sub-volcanic systems (Figure 4).~~ This structure would exist at depths beyond geodetic detection, at the base of the crust (ranging from 12 km at Fernandina, to 18 km at Sierra Negra [19]). We consider the simplest explanation for correlations in Galápagos displacements to be temporally variable melt supply from the plume into an asthenospheric melt layer, that causes neighbouring sub-volcanic reservoirs to simultaneously pressurise via pore-pressure diffusion. A variable melt supply would explain our observations of correlations in shallow magmatic reservoir pressure that are dominantly positive, but with some modifications due to (1) the impact of eruptions and resulting shallow stress changes on individual sub-volcanic systems and (2) variations in partitioning of melt entering the shallow crust due to differences in ascent mechanisms (Figure 4).
- o Lines 405-406 (Now Line 452): ~~occur due to a change in the partitioning of melt supply.~~ also be explained by pore pressure diffusion, where increased magma supply to one volcano creates a pressure gradient throughout the asthenospheric melt layer.
- o Lines 413–417 (Now Line 465): ~~Although the greatest volumes were still channeled to Sierra Negra (Figure 4), this intrusive supply variation suggests that the partitioning of magma between Galapagos volcanoes is dynamic (Figure 3d). Evidence of dynamic partitioning between these two volcanoes is also found during the~~ This negative correlation is also seen during the
- o Line 431-440 (Now Line 483): ~~Melt flux from the plume varies temporally, flushing through different sub-volcanic systems during eruptions [5], allowing subsiding volcanoes to resurge [44], and creating periods of elevated eruptive activity (e.g. 2005–2009, 2015–2022). The shallowest sub-volcanic reservoirs at each volcano in the Western Galápagos primarily change in pressure in response to this dynamic melt flux, causing positive correlations in displacements measured at the Earth's surface. Partitioning of melt between sub-volcanic systems exerts a secondary influence on decadal inter-eruptive deformation correlations and occurs at depths beyond geodetic or seismic detection.~~ This dynamic supply causes dynamic pore pressure changes throughout an asthenospheric melt layer, varying supply to the shallow

crust. This melt layer is likely composed of geochemically distinct sills, allowing dynamic stress transfer, while preserving isotopic homogeneity between volcanoes. Melt flux from the plume varies temporally, flushing through different sub-volcanic systems during eruptions [5], allowing subsiding volcanoes to resurge [44], and creating periods of elevated eruptive activity (e.g. 2005–2009, 2015–2022). However, magma is not supplied equally to each Galápagos volcano, with Sierra Negra receiving >50% of the total supply from 2000–2022 (Figure 4). Subsidence (indicative of limited magma supply to the shallow crust) has been observed twice at Sierra Negra, between 2000–2002 [47], and 2011–2012 [6], in both instances during a quiescent period of decreased eruptive activity across the Galapagos. At Kīlauea, local compositional heterogeneities at the plume alter the degree of partial melting, affecting magma supply to the shallow crust and the eruption rate [50]. Therefore, we speculate that variations in magma supply at Sierra Negra, caused by local plume heterogeneities, are primarily responsible for driving dynamic stress through an interconnected asthenospheric sill complex. Supply variations to other Galápagos volcanoes will also affect pore pressure stresses, though to a lesser extent than at Sierra Negra, as will

- o Lines 445–450 (Now Line 563): ~~In Hawai'i, correlations in activity and geophysical parameters have been attributed to a surge in magma supply that caused increased inflation and seismicity at Kīlauea between 2003–2007 [3]. During this melt supply surge, Mauna Loa switched from sustained deflation to inflation [3, 33], which has been attributed to pore pressure diffusion in an asthenospheric magma reservoir [34].~~, as has been discussed in Hawai'i.
- o Line 451 (Now Line 596): ~~In contrast, we~~ We
- o Line 454–454 (Now Line 597): ~~promotes a mode of melt ascent along established magmatic conduits, that is modified by dyking (discrete magma batches) during irregular supply partitioning.~~ is underlain by melt layer, comprising geochemically distinct sills, each sampling a different part of the Galápagos plume, allowing stress interactions at the point of supply to sub-volcanic magmatic zones (Figure 5).
- o Line 503 (Now Line 549): ~~partitioning.~~ supply.

2. *In figure 4 of this study, the authors present a model where there is a large magma body at depth feeding different sub-volcanic systems. Can the authors comment on the nature of this body, different regions of which must be fed from different mantle source components. Do you believe that this is a truly connected magma body that can share magma freely between systems (seemingly implausible with the geochemistry), is it an interconnected mush zone with limited lateral transport of melt, or is it a series of geochemically distinct sills, each of which might supply a different volcanic structure (this idea might have interest given the recent seismic observations of deep sill networks between Mauna Loa and Kīlauea on Hawai'i; <https://doi.org/10.1126/science.ade5755>). On a similar note, each volcano samples a different portion of the Galapagos mantle plume, would we necessarily expect there to be synchronous magma surges across this geochemically variable plume structure?*
 - o Figure 5 has now been included to illustrate the mechanism that we now propose in the text, allowing correlated deformation while preserving geochemical heterogeneity

3. *In summary, I'd like to see a greater discussion on how the correlated deformation signals can be reconciled with the fact that petrological evidence indicates that these systems are mostly isolated.*
 - o We reconcile our observations of correlated deformation signals with petrological evidence of isolated systems by suggesting that volcanoes of the Western Galápagos are underlain by an asthenospheric melt layer that through which dynamic stresses, driven by changes in magma flux, cause correlated deformation. Such a layer has been hypothesised in Hawai'i since 2012, and recent seismic studies (we thank the reviewer for bringing this to our attention), suggest that there is such a sill complex at depths of 36–43 km. By invoking such a complex at the base of the Galápagos crust, we believe we can account for correlated deformation, while maintaining geochemical heterogeneity

REVIEWERS' COMMENTS

Reviewer #1 (Remarks to the Author):

The authors have done a nice job of responding to suggestions. Acknowledging the different chemistry of the lavas from the various volcanoes is an important improvement to the paper. The authors now recognize in the text that the correlation of inflationary signals at volcanoes is not noteworthy. The paper now focuses more on shorter term changes including eruptions. The paper should be published, however there are two points that should be considered.

1) The focus on short term fluctuations is good, however some of the change points in Figures S7 and S8 are not at all obvious. The change points miss eruptions at Sierra Negra and Fernandina where signal went from inflation to deflation, which is very prominent. On the other hand, change points are found in areas where the signal is nearly flat. This is very puzzling and should be clarified.

2) The other point relates to the newly proposed mechanism for the observations. The problem I see with a model involving pressure transfer through an asthenospheric melt layer is the time scale of diffusion relative to the nearly simultaneous changes at distant volcanoes. While I understand the paper is observational I think it would be appropriate to point out that future modeling studies will need to explain the short time scales involved.

Paul Segall

Reviewer #2 (Remarks to the Author):

I do not agree with every interpretation in the revised manuscript, but the authors have done an excellent job of addressing reviewers' concerns via the revisions.

Reviewer #3 (Remarks to the Author):

In the initial version of this manuscript I expressed concerns that the ideas/models proposed by the authors were inconsistent with the geochemical diversity of the Galapagos volcanoes. The authors have taken some significant steps to address these concerns. As a geochemist, without the specific knowledge to address the plausibility of their model from a dynamical/physical standpoint, I am happy that their hypothesis could be consistent with the geochemical data. As a result, I have no further comments on this manuscript and suggest publication.

We thank the reviewers for acknowledging the changes that we have made to our manuscript. We respond to the current remarks below.

Reviewer #1 (Remarks to the Author):

The authors have done a nice job of responding to suggestions. Acknowledging the different chemistry of the lavas from the various volcanoes is an important improvement to the paper. The authors now recognize in the text that the correlation of inflationary signals at volcanoes is not noteworthy. The paper now focuses more on shorter term changes including eruptions. The paper should be published, however there are two points that should be considered.

1) The focus on short term fluctuations is good, however some of the change points in Figures S7 and S8 are not at all obvious. The change points miss eruptions at Sierra Negra and Fernandina where signal went from inflation to deflation, which is very prominent. On the other hand, change points are found in areas where the signal is nearly flat. This is very puzzling and should be clarified.

- We are fully aligned with the reviewers perspective on this. One of the most challenging parts of this paper was trying to determine the change points in the time series that are so evident to the human eye, using an independent method. We felt that the change point analysis approach detailed in the previous paper draft did a reasonable job, but still fails to exactly identify each point. This method has now been removed from the paper, and change points in figures S7 and S8 are now approximated visually (if there is a clear change in direction or rate of deformation that persists for weeks to months).
- Line 171: “use a change point algorithm” “identify change points in the time series on the basis of displacement direction or rate”
- Line 174: (Section 6)
- Line 175: “5” “4”
- Line 178: “0.3%” “0.21%”
- Line 494: “Change Point Analysis

As a test of statistical significance, we calculate the probability of simultaneous changes in deformation randomly occurring at multiple volcanoes by assuming a Poisson distribution (Equation 2), where X is the random variable, k is the number of times an event occurs and λ is the average number of times the event occurs (e.g. the number of turning points in a time series). *equation (2)*

To estimate λ , we fit segmented linear regression lines to our data, so that each segment is separated by a change point (PELT off-line change point detection algorithm, in the ruptures Python library [47]). The number and location of change points is determined using a cost function, alongside a penalty value to avoid over-fitting (Figures S7, S8). We take the average number of change points across all time series as $\lambda = 5$, and use this to estimate the probability of multiple volcanoes showing changes in deformation rate at the same time. Similarly, we estimate the probability of eruptions randomly occurring at the same time using Equation 2, using average eruption rates at each volcano between 1911–2023 (1911 was the first recorded 20th eruption).”

2) The other point relates to the newly proposed mechanism for the observations. The problem I see with a model involving pressure transfer through an asthenospheric melt layer is the time scale of diffusion relative to the nearly simultaneous changes at distant volcanoes. While I understand the paper is observational I think it would be appropriate to point out that future modeling studies will need to explain the short time scales involved.

- The reviewer makes a good point, and we have added the following statement in text:
- Line 362: “An aspect that remains unclear are the timescales across which this process occurs, and how well they are resolved by Sentinel-1 InSAR (return period of 6–12 days). Future modelling studies are required to fully investigate this”.
- It is important to bear in mind that while correlated behaviour may appear simultaneous, we can only resolve these changes with a maximum resolution of 6 days (sentinel-1 return period).

Reviewer #2 (Remarks to the Author):

I do not agree with every interpretation in the revised manuscript, but the authors have done an excellent job of addressing reviewers' concerns via the revisions.

Reviewer #3 (Remarks to the Author):

In the initial version of this manuscript I expressed concerns that the ideas/models proposed by the authors were inconsistent with the geochemical diversity of the Galapagos volcanoes. The authors have taken some significant steps to address these concerns. As a geochemist, without the specific knowledge to address the plausibility of their model from a dynamical/physical standpoint, I am happy that their hypothesis could be consistent with the geochemical data. As a result, I have no further comments on this manuscript and suggest publication.

- We again thank both Reviewer #2 and Reviewer #3 for their comments on our initial manuscript submission. The comments and literature suggestions they provided were critical in allowing us to develop our understanding of the processes driving correlated deformation in the Galápagos. We have softened our language regarding our conceptual model, to allow it to be interpreted and adapted by future studies.
- Line 253: "would" "could"
- Line 344: "analogous to Hawaii"
- Line 345: "causes" "may cause"